# First Report and Comparative Genomic Analysis of *Mycoplasma capricolum* subsp. *capricolum* HN-B in Hainan Island, China

**DOI:** 10.3390/microorganisms10112298

**Published:** 2022-11-19

**Authors:** Zhenxing Zhang, Meirong He, Junming Jiang, Xubo Li, Haoyang Li, Wencan Zhang, Si Chen, Li Du, Churiga Man, Qiaoling Chen, Hongyan Gao, Fengyang Wang

**Affiliations:** Hainan Key Laboratory of Tropical Animal Reproduction & Breeding and Epidemic Disease Research, Engineering Key Laboratory of Haikou, College of Animal Science and Technology, Hainan University, Haikou 570228, China

**Keywords:** *Mycoplasma capricolum* subsp. *capricolum*, *Mycoplasma mycoides* cluster, gene islands, virulence factor, Hainan Island, comparative genomics

## Abstract

*Mycoplasma capricolum* subsp. *Capricolum* (Mcc) is an important member of the *Mycoplasma mycoides* cluster (Mm cluster) and causes caprine contagious agalactia. Mcc can infect goats of all age groups, especially pregnant ewes and kids. It can cause the abortion in pregnant ewes and the death of goat kids, leading to enormous losses in the goat breeding industry. To date, the prevalence of epidemic Mcc strains on Hainan Island, China, remains unclear. This study aimed to isolate and identify Mcc strains endemic to Hainan Island, China. Genome sequencing and comparative genomic analysis were performed to reveal the molecular characteristics and evolutionary relationships of the isolated strain. Mcc HN-B was isolated and identified in Hainan Island, China. The Mcc HN-B genome consists of a 1,117,925 bp circular chromosome with a 23.79% G + C content. It contains 912 encoding genes, 3 gene islands, and 14 potential virulence genes. The core genome with the features of the Mm cluster and the specific genes of Mcc HN-B were identified by comparative genomic analysis. These results revealed the evolutionary relationship between Mcc HN-B and other members of the Mm cluster. Our findings provide a reference for further studies on the pathogenic mechanism and local vaccine development of Mcc.

## 1. Introduction

The *Mycoplasma mycoides* cluster (Mm cluster) is a group of several mycoplasmas that have close genetic relationships and the same pathogenic characteristics [1,2]. It includes pathogenic mycoplasmas, such as *Mycoplasma mycoides* subsp. *capri* (Mmc), *Mycoplasma mycoides* subsp. *mycoides* large colony (MmmLC), *Mycoplasma capricolum* subsp. *capricolum* (Mcc), *Mycoplasma capricolum* subsp. *capripneumoniae* (Mccp), *Mycoplasma mycoides* subsp. *mycoides* small colony (MmmSC), and *Mycoplasma leachii* (Ml) [3]. Mccp is the causative agent of contagious caprine pleuropneumonia, which can lead to huge economic losses in the goat breeding industry [4,5]. MmmSC can cause contagious bovine pleuropneumonia, presenting an enormous threat to the cattle industry [6]. Mcc, Mmc, and MmmLC can cause mastitis, arthritis, keratoconjunctivitis, pneumonia, and septicemia (MAKePS) syndrome in ruminants and are also the causative agents of contagious agalactia [4,7]. In addition, Mcc is highly virulent and pathogenic in goats. It can infect goats of all age groups and cause abortion of ewes and death of cubs [8]. Nicolas et al. [9] found that MmmLC and Mcc can be interspecifically transmitted between different taxa of ruminants, which may cause great losses in the breeding industry.

With the development of bacterial isolation and culture and molecular diagnostic techniques, Mcc has been found in many countries and regions, such as Spain, Portugal, and the United States [8,10,11]. Researchers have used various methods to study Mcc to prevent and treat Mcc-related diseases. Using PCR and restriction enzyme digestion technologies, Rodriguez et al. [12] established an approach to differentiate Mcc from other members of the Mm cluster. Based on the suppression subtractive hybridization technology, Maigre et al. [13] developed a specific PCR diagnostic method for Mcc. In addition, comparative genomic analysis has been performed to reveal the genetic relationship and variation between different strains of Mcc and between Mcc and other mycoplasmas [14,15]. Moreover, antimicrobial susceptibility tests of Mcc have been carried out, and further studies have investigated its drug resistance mechanisms [16,17].

To date, a total of five Mcc genomes can be found in the National Center for Biotechnology Information (NCBI, 24 September 2022), yet only one genome has been completely assembled. The complete genome of Mcc-endemic strains is a fundamental prerequisite for studying their molecular characteristics and pathogenic mechanisms. It has an important reference value for Mcc evolutionary relationship research, epidemiological investigations, and regional vaccine development.

In the present study, Mcc HN-B was isolated and identified for the first time on Hainan Island, China. We obtained a high-quality Mcc HN-B genome using whole-genome sequencing. All members of the Mm cluster (Mmc HN-A, MmmLC 95010, Mcc ATCC 27343, Mccp M1601, MmmSC PG1, and Ml PG50) were selected to perform comparative genomic analysis with Mcc HN-B. This study provides a valuable reference for acquiring strain information on Mcc, analyzing the molecular characteristics of the Mcc HN-B genome in detail, and revealing the genetic evolutionary relationships among all members of the Mm cluster. In addition, it will help foster regional prevention, pathogenic mechanism research, and drug development for Mcc.

## 2. Materials and Methods

### 2.1. Isolation and Identification of Mcc HN-B

In a black goat breeding factory, a two-month-old female lamb suffered from elbow swelling of the left forelimb, limp, ataxia, and other symptoms. A sterile syringe was used to draw the liquid inside the cyst of the elbow joint to isolate and identify the bacteria. The volume of the joint fluid was approximately 100 µL, which was clear with a light-yellow clot. It was stored by the Hainan Key Laboratory of Tropical Animal Breeding and Disease Research.

We added 50 µL of joint fluid to 450 µL of sterile PBS and mixed thoroughly. This procedure was repeated thrice. Then, different concentrations of 100 µL of articular fluid were inoculated into 10 mL of mycoplasma liquid medium, which included 0.03 g glucose, 0.025 g yeast powder, 0.02 g sodium pyruvate, 0.21 g pleuropneumonia-like organism (PPLO) broth (Difco, Tucker, GA, USA), 8000 IU penicillin G (BioFroxx, Einhausen, Germany), 2 mL horse serum (Solarbio, Beijing, China), and 25 µL of 0.4% phenol red solution (Amresco, Boise, ID, USA). In addition, 1.5% agar powder (BioFroxx, Einhausen, Germany) was added to the mycoplasma liquid medium to prepare a mycoplasma solid medium. After inoculation, the medium was placed in a cell incubator (37 °C and 5% CO_2_) for culturing.

Changes in the mycoplasma liquid and solid media were observed and recorded. When the liquid medium became yellow or cloudy, the medium was filtered through a filter with a pore size of 0.45 μm. The filtered medium was seeded and subcultured in fresh medium at 10% volume. Purified bacteria were obtained by repeated passaging (three times). The inoculated mycoplasma solid medium was observed daily under an ordinary light microscope. Once suspicious colonies were found, they were cloned and purified on new solid medium over time. Giemsa staining was performed, and the cells were observed and photographed under a microscope.

The synthetic universal primers for the bacterial *16S rRNA* gene [18] were F, 5′-AGAGTTTGATCCTGGCTCAG-3′ and R, 5′-GGTTACCTTGTTACGACTT-3′. Purified colonies were selected for amplification and identification of the *16S rRNA* gene. The PCR system included upstream and downstream primers (20 pmol each), ddH_2_O (20 µL), 2× Taq PCR Mix (25 µL), and 1 µL of DNA template. The PCR procedure involved predenaturation at 94 °C for 5 min, denaturation at 94 °C for 30 s, annealing at 56 °C for 50 s, extension at 72 °C for 90 s, 32 cycles of circulation, and termination of extension at 72 °C for 10 min.

### 2.2. Mcc HN-B Strain Culture and Genomic DNA Extraction

Purified colonies were selected and inoculated into 100 mL of mycoplasma liquid medium for expanded culture. The genome of Mcc HN-B was extracted using a HiPure Bacterial DNA Kit (Magen, Guangzhou, China). Qubit (Thermo Fisher Scientific, Waltham, MA, USA) and NanoDrop (Thermo Fisher Scientific, Waltham, MA, USA) were used to test the DNA quality.

### 2.3. ONT Sequencing and Illumina Sequencing

After the DNA quality test, DNA libraries for Oxford Nanopore Technology (ONT) sequencing and Illumina sequencing were constructed. First, according to the protocol offered by ONT, the genomic DNA was broken into segments averaging approximately 10 kB using G-tubes (Covaris, Woburn, MA, USA). Next, NEBNext End Repair/dA-Tailing Module reagents (E7546, NEB, USA) and the NEBNext FFPE DNA Repair kit (M6630, NEB, USA) were used for end repair. NEBNext Quick Ligation Module reagents (E6056, NEB, USA) were used for joint connection. Finally, we used the AMPure XP system (Beckman Coulter, Brea, CA, USA) to purify the DNA library and performed sequencing using the long-read sequencing platform Oxford Nanopore PromethION (Oxford Nanopore Technologies, Oxford, UK).

In accordance with the instructions of the NEBNext^®^ ΜLtra™ DNA Library Prep Kit for Illumina (NEB, USA), the genomic DNA segmented by random ultrasound was subjected to end repair, polyA tail processing, and adaptor ligation. Then, the PCR and AMPure XP systems (Beckman Coulter, Brea, CA, USA) were employed for enrichment and purification of the library. After qualification testing using a 2100 Bioanalyzer (Agilent, Santa Clara, CA, USA), paired-end technology (PE 150) was applied for sequencing on an Illumina NovaSeq 6000 sequencer.

### 2.4. Genome Assembly and Genome Component Annotation

Flye (version 2.8.1-b1676) [19] was used to acquire long reads from the ONT sequencing and assemble from scratch. FASTP (version 0.20.0) [20] was used to filter the original data from the Illumina platform. Finally, Pilon (version 1.23) [21] was used to correct the genome sequence via filtered reads and obtain the final genome sequence.

The Mcc HN-B composition prediction involves open reading frames (ORFs), rRNAs, tRNAs, sRNAs, gene islands (GIs), clustered regularly interspaced short palindromic repeats (CRISPR), transposons, interspersed repeat elements, tandems, and prophages. They were predicted using the NCBI prokaryotic genome annotation pipeline [22], RNAmmer (version 1.2) [23], tRNAscan (version 1.3.1) [24], cmscan (version 1.1.2) [25], IslandPath-DIMOB (version 1.0.0) [26], CRISPRFinder (4.2.17) [27], TransposonPSI (version: 20100822) [28], RepeatMasker (version 4.0.5) [29], TRF (version 4.09) [30], and PHAST (version 2.0) [31], respectively.

### 2.5. Gene Function Analysis

Based on the screening standard of E-value < 1 × 10^−5^, databases, such as non-redundant protein sequence database (Nr), UniProt/Swiss-Prot, Kyoto Encyclopedia of Genes and Genomes (KEGG), Clusters of Orthologous Groups (COG), Carbohydrate-Active EnZYmes database (CAZy), Transporter Classification Database (TCDB), Pathogen–Host Interactions database (PHI), and Virulence Factors of Pathogenic Bacteria database (VFDB), were used to analyze the genomic sequence of Mcc HN-B in a comparative manner and annotate the gene function. According to the default software parameters, the Gene Ontology database (GO), Pfam database (version 32.0) and Comprehensive Antibiotic Resistance Database (CARD) were used to predict DNA functions. The software information is listed in Appendix A.

In addition, secreted and transmembrane proteins were predicted using SignalP4.0 [32] and TMHMM (version 2.0) [33], respectively. Type three secretion systems (T3SSs) and secondary metabolic gene clusters were predicted using EffectiveT3 (version 1.0.1) [34] and antiSMASH (version 4.1.0) [35]. We also predicted the two-component systems (TCSs) of Mcc HN-B.

### 2.6. Collinearity Analysis and SNP/InDel/SV Statistics

As the target genome, Mcc HN-B belongs to one of the members of the Mm cluster. The six genomes selected were different subspecies within the Mm cluster. First, MUMmer (version 3.1) [36] was used to compare the target and reference genomes to identify the large-scale collinearity between genomes. Subsequently, SyRI (version 1.4) [37] was applied to test the local position arrangement of the samples.

Single-nucleotide polymorphisms (SNPs) and insertions–deletions (InDels) were tested and analyzed using MUMmer (version 3.1). Structural variation (SV) was examined and analyzed using SyRI (version 1.4).

### 2.7. Gene Family Analysis

In accordance with the bidirectional best-hit standard (80% of the shortest protein sequence has 40% amino acid similarity), gene family analysis was performed on the target and reference genomes. Diamond [38] (version 2.0.7) and OrthoMCL (version 1.4) [39] were used to perform a comparison and similarity clustering of their amino acid (or nucleotide) sequences. Finally, homologous gene clusters and species distributions of each protein cluster were obtained.

### 2.8. Core/Pan-Genome Analysis

The pan-genome includes both core and dispensable genomes. Among them, the genes present in all strains constitute the core genome, and genes excluding these common genes constitute the dispensable genome. In dispensable genomes, genes that are unique to only one strain are named ‘unique genes’; a gene that is shared by more than one strain, but not all, is named an ‘accessory gene’.

Core/pan-genome analysis was performed using Mcc HN-B and the six reference genomes. Strain-specific genes of Mcc HN-B and reference strains were counted according to the gene family clustering results in Section 2.7. Strain-specific genes included genes in strain-specific gene families and genes that were not involved in the above clustering.

### 2.9. Phylogenetic Tree Analysis and ANI Analysis

Considering the homologous gene cluster analysis results of Mcc HN-B and the reference genomes, we selected homologous genes with single copies. Multiple sequence alignment and alignment quality control were performed using MUSCLE [40] (version 3.8.31) and Gblocks [41] (version 0.91B), respectively. IQ-Tree [42] (version 1.6.3) was used to construct phylogenetic trees according to the maximum likelihood method.

The average nucleotide identity (ANI) value refers to the average base similarity between homologous segments of two microbial genomes. Normally, an ANI value of 95% is used as the classification threshold to distinguish between different species, which shows a high degree of discrimination among closely related species. We used Pyani [43] (version 0.2.7) to calculate the ANI value of the alignment region between the Mcc HN-B genome and each reference genome.

### 2.10. GenBank Accession Number

The annotated whole-genome sequence of the Mcc HN-B strain was submitted to the GenBank database under accession number CP101903.

## 3. Results and Discussion

### 3.1. Results of Mcc HN-B Isolation and Identification

The joint fluid was cultured in mycoplasma liquid medium. After 17 h, the color of the liquid medium changed from red to yellow. After 25 h, the liquid medium became turbid and precipitated. After 48 h, the color of the liquid medium did not change, and the precipitation reached its maximum. Simultaneously, the joint fluid was inoculated into the mycoplasma solid medium. Small transparent colonies began to appear 24 h after inoculation. After 36 h, the colonies continued to grow, and the color became milky white. After three subcultures, we successfully isolated and purified the mycoplasma strain.

Under the low-power microscope, the colonies were mostly round, and the central umbilicuses were obvious (Figure 1A). Single colonies of this strain were selected for Giemsa staining and observed under magnification of 1000 times (Figure 1B); purple particles were observed in the microscopic field, and the isolated strains were spherical or arc shaped.

The *16S rRNA* gene of the isolated strain was successfully amplified and sequenced (Figure 1C). The NCBI analysis results showed that the *16S rRNA* gene sequence of the isolated strain had the highest homology (99.71%) with Mcc 14DD0024. The strain was identified as Mcc by culture characteristics, morphological characteristics, and PCR amplification, and named “Mcc HN-B.”

### 3.2. General Characteristics of the Mcc HN-B Genome

To ensure the accuracy and reliability of the subsequent information analysis, the original data generated by the ONT and Illumina sequencing platforms were filtered and processed to obtain valid data. The Illumina sequencing data and ONT sequencing data of Mcc HN-B were 1.6 Gb (~1428.57×) and 1.12 Gb (~1000×), respectively. Through genome assembly and correction, the complete genomic sequence of Mcc HN-B, with a total length of 1,117,925 bp, was obtained. Combined with the prediction results of the coding genes, the Circos software (version 0.69–9) was used to draw the genome circle map of Mcc HN-B to comprehensively display its genome characteristics (Figure 2).

The whole genome of the Mcc HN-B strain with a 23.79% G + C content includes 912 coding genes, 3 GIs, 175 tandem repeats, and 1 DNA transposon (Table 1). We first predicted six potential CRISPRs in Mcc HN-B (Appendix A), which require further validation. Additionally, 912 proteins were encoded by the Mcc HN-B genome, including 12 transmembrane proteins (Appendix A) and 46 secreted proteins (Appendix A). The results of the functional annotation of the Mcc HN-B genome are shown in Appendix A.

#### 3.2.1. GIs

Three GIs were first identified in the Mcc HN-B genome. Among them, 9, 32, and 26 genes were included in GI1 (GI_1), GI2 (GI_2), and GI3 (GI_3), respectively (Figure 3, Appendix A).

On GI_1, *NO343_00050*, *NO343_00060*, and *NO343_00070* encode three membrane proteins; *NO343_00075* encodes one lipoprotein; and *NO343_00055*, *NO343_00080*, and *NO343_00085* encode three hypothetical proteins. The DUF285 family protein encoded by *NO343_00055* may be related to the adhesion of Mcc HN-B. *NO343_00065* encodes a DnaJ-domain-containing protein [44]. It could act as an auxiliary molecular chaperone of heat shock protein 70 (HSP70) to participate in different physiological processes. *NO343_00090* encodes a DDE transposase, which may provide a basis for inserting sequence transposition activity and promoting the evolution and environmental adaptation of bacteria [45].

GI_2 and GI_3 had similar compositions. Among them, 20 genes encode proteins that are structurally and functionally similar and arranged in highly similar positions. These genes encode hypothetical proteins, membrane proteins, and lipoproteins that are related to the virulence of Mcc HN-B. *NO343_02020* in GI_2 and *NO343_04000* in GI_3 encode transposases of the IS3 family. We speculated that GI_2 and GI_3 have undergone gene transfer in related species or the same species as Mcc, eventually leading to their similar structure and genetic composition.

Additionally, compared with GI_3, GI_2 encodes more proteins, including three ATP-binding proteins (NO343_02160, NO343_02165, and NO343_02170), two membrane proteins (NO343_02145 and NO343_02150), three hypothetical proteins (NO343_02135, NO343_02155, and NO343_02175), and one TIR-domain-containing protein (NO343_02015). Comparatively, GI_3 encodes an extra hypothetical protein (NO343_04030) and two membrane proteins (NO343_04020 and NO343_04025).

#### 3.2.2. Virulence Factors

VFDB is a database dedicated to the study of pathogenic factors in bacteria, chlamydia, and mycoplasma. Fourteen genes in the Mcc HN-B genome were annotated to the VFDB database. Among them, six and three genes were annotated to surface lipoproteins and capsules, respectively (Appendix A). *NO343_00170*, *NO343_03810*, *NO343_04655*, *NO343_04660*, *NO343_04665*, and *NO343_04670* encode six Vmm proteins, which are all surface lipoproteins, contributing to high-frequency phase variation in immunodominant antigens. The proteins encoded by *GalU* (*NO343_02230*), *OppF* (*NO343_02495*), and *MSC_0991* (*NO343_02500*) belong to the capsule. Only one gene was annotated to GAPDH, hemolysin, elongation factor thermal unstable (EF-Tu), streptococcal enolase, or pyruvate dehydrogenase E1 beta subunit (PDH-B).

*gapA* (*NO343_00195*) encodes the GAPDH protein of Mcc HN-B. Recent studies [46] have indicated that the GAPDH protein is not only a key enzyme in the glycolytic metabolic pathway but also involved in intracellular digestion, DNA repair, apoptosis, and other life processes. Hoelzle et al. [47] reported that the MSG1 protein with GAPDH activity in *Mycoplasma suis* (*M. suis*) also functions in adhesion to porcine erythrocytes. We speculated that NO343_00195 plays an adhesive role in the Mcc HN-B strain to enhance Mcc infection in the host.

Hemolysin is a common exotoxin that can cause the dissolution of red blood cells, leading to capillary necrosis [48]. *hlyA* often exists in *Listeria monocytogenes* [49] and *Vibrio cholerae* [50]. Additionally, some pathogenic mycoplasma genomes contain genes encoding hemolysin A, such as *Mycoplasma hyopneumoniae* (*M. hyopneumoniae*), *Mycoplasma ovipneumoniae*, Mccp [51], and *Mycoplasma conjunctivae*. In the present research, we found that *hlyA* (*NO343_02185*) also exists in the Mcc HN-B genome, with a sequence homology rate of 98.02% compared with *hlyA* (*MCAP_0055*) in the Mcc ATCC 27343 genome submitted in 2005.

PDHB is an important component of pyruvate dehydrogenase. EF-Tu is an indispensable functional protein for life activities, accounting for approximately 10% of the total protein of *Mycoplasma pneumoniae* (*M. pneumoniae*) [52]. Pinto et al. [53] identified EF-Tu and PDHB as highly antigenic proteins of *M. hyopneumoniae* by Western blotting, which can be recognized by the host immune system. Dallo et al. [54] demonstrated that EF-Tu and PDH-B, two surface proteins of *M. pneumoniae*, could bind to fibronectin, which is a component of extracellular matrix, and affect mycoplasma adherence to the host and infection [55]. *pdhB* (*NO343_02990*) and *tuf* (*NO343_02670*) were identified in the Mcc HN-B genome, which might also facilitate the adhesion and infection of Mcc to the host.

α-enolase exists on the surface of many bacteria, has good immunogenicity in *M. suis*, and can enhance the adhesion ability of bacteria to the host [56]. Esgleas et al. showed that α-enolase is a surface protein with fibronectin-binding activity that plays a role in bacterial adhesion and invasion [57,58]. The potential virulence gene *eno* (*NO343_02925*) was also identified in Mcc HN-B, which may be important for Mcc to invade the host.

We also found that *NO343_03060* and *NO343_03295* encode two kinds of peptidase S41 in the Mcc HN-B genome. A study demonstrated [59] that S41 peptidase is closely related to the proteolytic phenotype of Mmc. Deletion of the S41 peptidase can cause the phenotypic change in Mmc and increase the production of hydrogen peroxide. Therefore, NO343_03060 and NO343_03295 may be vital for the proteolytic phenotype and virulence sensitivity of Mcc HN-B.

Finally, a type I restriction modification (RM) system was identified in the Mcc HN-B genome. HsdR is encoded by *NO343_01990*, HsdM by *NO343_01975*, and HsdS by *hsdS* (*NO343_01985*, *NO343_02000*, *NO343_04805*, *NO343_04810*). This system contains repeated *hsdS* genes, and Mcc HN-B may change its phenotype through phase variation [60]. Meanwhile, the type I RM system may influence the virulence of Mcc HN-B by regulating the expression of virulence genes [61,62]. *NO343_01175* and *NO343_03390* of Mcc HN-B encoding two type II RM system-related proteins were also identified.

#### 3.2.3. Metabolism

The histidine phosphocarrier protein (HPr) is a component of the phosphotransferase system (PTS). Kundig [63] et al. reported its function in hexose phosphorylation in 1964. The PTS system has been found in an increasing number of fungi, archaea, and bacteria. Most of the known PTS systems have similar structures, consisting of the enzyme I complex (EI), HPr, and enzyme II complex (EIIA, EIIB, EIIC, EIID). The main function of the PTS system is to mediate the absorption and phosphorylation of carbohydrates and to participate in the regulation of carbon metabolism and bacterial virulence [64].

In this study, we identified 20 Mcc HN-B genes involved in the PTS system. Two cytosolic phosphotransferases, EI and HPr, are encoded by *NO343_03025* and *NO343_00615*, respectively. *NO343_03030* encodes EIIA^Glc^. *NO343_01255*, *NO343_01260*, *NO343_01275*, *NO343_01560*, and *NO343_02375* encode EIICB^Glc^. *NO343_01870* and *NO343_03925* encode EIIA^Mtl^. *NO343_01860* and *NO343_03920* encode EIICB^Mtl^. *NO343_01610* encodes EIIC^Lev^. *NO343_04645* encodes an ascorbate PTS system EIIB component. *NO343_04650* encodes the ascorbate PTS system EIIA or EIIAB component. *NO343_02540* encodes the *N*-acetylglucosamine PTS system EIICBA or EIICB component. *NO343_02675* encodes the sucrose PTS system EIIBCA or EIIBC component. *NO343_04795* encodes the sucrose PTS system EIIBCA or EIIBC component. *NO343_01605* and *NO343_04800* encode 1-phosphofructokinase.

Ten different types of EII were identified in Mcc HN-B. These enzymes enable the PTS system of Mcc HN-B to transport a wide range of carbohydrates. The major glucose transport system in Mcc HN-B was the glucose PTS system. Mcc also contained mannitol PTS, *N*-acetylglucosamine PTS, sucrose PTS, and fructose PTS. Studies have shown that *N*-acetylglucosamine PTS is more effective in promoting glucose transport and utilization than maltose PTS and β-glucoside PTS in the absence of glucose PTS, mannose PTS, and glucokinase [65]. Christine et al. proved that when fructose is the only carbon source, the fructose PTS of *Clostridium acetobutylicum* becomes the main fructose uptake system [66]. Combined with the PTS characteristics of Mcc HN-B, we hypothesized that the *N*-acetyl glucosamine PTS could act as a substitute to promote glucose transport when Mcc HN-B lacks glucose PTS and glucokinase. Moreover, the presence of fructose PTS in Mcc HN-B may make it possible to use fructose as a carbon source, thus enhancing the environmental stress resistance of the strain.

Additionally, 33 Mcc HN-B genes were involved in the metabolic process of the ATP-binding cassette (ABC) transporters, including a general nucleoside transport system, a phosphonate transport system, and an oligopeptide transport system. Twenty-seven Mcc HN-B genes were involved in the glycolysis/gluconeogenesis metabolic pathway. The metabolic pathways of glucose and pyruvate were intact, which may allow Mcc to generate energy under the hypoxic conditions.

### 3.3. Results of Comparative Genomic Analysis

#### 3.3.1. Collinearity Analysis Results

We first confirmed the genomes of six strains (Mcc ATCC 27343, Mccp M1601, Mmc HN-A, MmmLC 95010, MmmSC PG1, and Ml PG50) in the Mm cluster as the target genomes (Table 2). The Mcc HN-B genome was used as the reference genome for collinearity analysis with target genomes.

The collinearity results were statistically analyzed (Appendix A). A plot of the parallel collinearity results was drawn based on these results (Figure 4). The relationships between Mcc ATCC 27343, Mccp M1601, and Mcc HN-B were the closest, followed by Ml PG50. MmmSC PG1 had the farthest genetic relationship with Mcc HN-B, followed by Mmc HN-A and MmmLC 95010. Many reverse-matching regions were found between the genomes of *Mycoplasma capricolum* and *Mycoplasma mycoides*. However, this phenomenon was not observed in Ml. Thus, the evolutionary distance between Ml and *Mycoplasma capricolum* genomes was closer than that between Ml and *Mycoplasma mycoides* genomes.

#### 3.3.2. Statistical Results of SNPs, InDels, and SVs

The number of SNPs, InDels, and SVs between Mcc HN-B and the target strains was determined and analyzed (Table 3). The figure of the variation types is shown (Appendix A).

As shown in Table 3, the genomes of Mcc HN-B and Mcc ATCC 27343 had the lowest numbers of SNPs, insertions, and deletions. In addition, there was one more SV between Mcc HN-B and Mcc ATCC 27343 than between Mcc HN-B and Ml PG50. Therefore, Mcc HN-B had the lowest genome variations when compared with Mcc ATCC 27343, followed by Mccp M1601. Combined with the picture of variation types (Appendix A), many similar inverted regions were found between *Mycoplasma capricolum* and *Mycoplasma mycoides*. These regions corresponded to the reverse-matching regions mentioned above. *Mycoplasma capricolum* and Ml were more similar in genome structure than *Mycoplasma mycoides*.

### 3.4. Molecular Characterization of the Mm cluster

The Mm cluster is a group of mycoplasmas that belong to the genus *Mycoplasma*. It includes six important pathogenic mycoplasmas for ruminants: Mmc, MmmLC, MmmSC, Mcc, Mccp, and Ml. Therefore, finding the core genome, homologous gene family, and other molecular characteristics of the Mm cluster is of great significance for rapid clinical diagnosis, vaccine development, and disease treatment.

The six selected target genomes were different subspecies within the Mm cluster. Together with the Mcc HN-B genome, we performed gene family, core/pan-genome, phylogenetic tree, and ANI analyses to determine the molecular characteristics of the Mm cluster.

#### 3.4.1. Gene Family Analysis and Core Genome Identification of the Mm Cluster

We first collected the results of gene family analysis for the seven mycoplasma strains (Table 4). A total of 580 orthologs of the Mm cluster and unique families in each strain were identified therein (Appendix A, Figure 5).

Subsequently, we identified the core/pan-genomes with the characteristics of the Mm cluster using core/pan-genome analysis. The core genome contained 580 orthologs, which were key components for maintaining the basic vital activities and molecular characteristics of the Mm cluster. Among them, 18 orthologs encode 13 lipoproteins, 3 putative lipoproteins, and 2 putative prolipoproteins. These lipoproteins include one LPPA family lipoprotein, one LPPD family lipoprotein, and one MOLPALP family lipoprotein. They constitute the basic virulence of the Mm cluster, which provides a reference for further exploration of their virulence and pathogenic mechanisms.

The pan-genome comprised 1363 orthologs, which showed genetic diversity within the Mm cluster. All shared genes and specific genes of each strain were obtained (Table 5). This provides a basis for identifying the molecular characteristics and evolutionary distances between different subspecies within the Mm cluster.

#### 3.4.2. Phylogenetic Tree and ANI Analysis

According to the orthologs’ clustering results of the Mm cluster in Section 3.4.1, a phylogenetic tree was constructed based on the single-copy orthologs (Figure 6). We found that Mcc HN-B had the closest relative genetic distance to Mccp M1601, followed by Ml PG50. This was consistent with the analysis results of the genomic structural variations of each strain in Section 3.3.2. Moreover, Mcc HN-B had the farthest relative genetic distance from Mmc HN-A and MmmLC 95010. Compared with the other subspecies of *Mycoplasma mycoides*, the relative genetic distance between MmmSC PG1 and Ml PG50 was smaller. They had a closer genetic relationship, and their hosts were both cattle.

Additionally, we discovered that Mcc HN-B had a smaller genetic distance from Mccp M1601 than Mcc ATCC 27343. Mccp M1601 was isolated from China in 2016, whereas Mcc ATCC 27343 was isolated from America in 2005. This result may be due to the geographic differences in Mcc and the consequences of the long-term evolution of these strains.

Finally, a heatmap based on ANI values was drawn to show the average base similarity of homologous fragments between the two strains (Figure 7). The ANI values among Mcc HN-B, Mccp M1601, and Mcc ATCC 27343 were all greater than 0.96. Comparatively, the ANI values were all lower than 0.95 between Ml PG50 and the three strains mentioned above. The ANI values between Mmc HN-A and MmmLC 95010 were greater than 0.95. The ANI value of either of them with MmmSC PG1 was lower than 0.95, which could effectively distinguish them. Mcc HN-B showed the highest ANI value with Mccp M1601, which is in accordance with the phylogenetic tree analysis results.

### 3.5. Molecular Characterization of the Mycoplasma Capricolum Genome

Mcc HN-B, Mccp M1601, and Mcc ATCC 27343 belong to *Mycoplasma capricolum*. Orthologs of the three genomes listed in Appendix A were selected for comparison. A Venn diagram was constructed to show the results (Figure 8).

Through comparative analysis, we identified 692 orthologs (Appendix A) and 1058 orthologs in the core genome and pan-genomes of *Mycoplasma capricolum*, respectively. In the core genome, 36 orthologs encode 24 lipoproteins, 2 prolipoproteins, 8 putative lipoproteins, and 2 putative prolipoproteins. Apart from the common LPPA and LPPD family lipoproteins, one VmcF lipoprotein, one PARCEL family lipoprotein, and three MOLPALP family lipoproteins were predicted to exist in *Mycoplasma capricolum*.

Compared with the lipoproteins in the core genome of the Mm cluster, the 18 extra lipoproteins in *Mycoplasma capricolum*, such as VmcF lipoprotein and PARCEL family lipoprotein, may enhance the virulence of *Mycoplasma capricolum*. Additionally, they could be markers of *Mycoplasma capricolum*. Moreover, specific genes in the genomes of Mcc HN-B, Mcc ATCC 27343, and Mccp M1601 were also found, which added support for their specific identification.

Finally, Mcc HN-B shared 767 orthologs with Mcc ATCC 27343, which is 75 more orthologs than those in the core genome of *Mycoplasma capricolum*. These 75 orthologs were specific to Mcc and were of great significance for distinguishing between Mcc and Mccp. Meanwhile, it is of particular concern that Mcc HN-B shares 721 orthologs with Mccp M1601, which is 29 more orthologs than those in the core genome of *Mycoplasma capricolum*. As both Mcc HN-B and Mccp M1601 were isolated from China, these 29 orthologs may have regional characteristics in China.

### 3.6. Identification of Caprine-Host-Specific Orthologs

The hosts of Mcc, Mccp, and Mmc are all goats. To explore the caprine-host-specific orthologs of these strains in China, the orthologs of the three strains (Mcc HN-B, Mccp M1601, and Mmc HN-A) isolated from China were selected for comparison (Appendix A). The results are shown in a Venn diagram (Figure 9).

The core and pan-genomes of the caprine-host strains contained 653 and 1133 orthologs, respectively (Figure 9). Caprine-host-specific orthologs were included in this core genome. Subsequently, MmmSC PG1 and Ml PG50, both with bovine hosts, were selected for further screening (Figure 9). The core and pan-genomes of the bovine-host strains had 695 and 1011 orthologs, respectively. Finally, the caprine-specific core genome was compared with the bovine-specific core genome to obtain the number of caprine-host- specific orthologs in China.

Finally, 63 orthologs were identified as caprine-host-specific orthologs in China (Appendix A). In total, 105 orthologs were identified as bovine-host-specific orthologs (Appendix A). Among the caprine-host-specific orthologs, 11 orthologs encode seven lipoproteins, one prolipoprotein, and three putative lipoproteins. Five orthologs encode three membrane proteins and two putative membrane proteins. The other orthologs encode proteins, such as M17 peptidase, ATP-binding cassette transporter permease subunit, type I RM system subunit M, and type I RM system subunit R. We speculated that these genes might play an important role in host recognition, virulence regulation, substance transportation, and energy generation when goats are infected with Mcc, Mccp, or Mmc. These results provide a reference for the further study of the specific infection mechanisms of Mcc, Mccp, and Mmc in goats.

## 4. Conclusions

In this study, the Mcc HN-B strain was successfully isolated and identified from Hainan Island, China, for the first time. According to whole-genome sequencing analysis, we first identified three GIs in the Mcc HN-B genome. At the same time, we analyzed the type I RM and PTS systems of Mcc HN-B, revealing 14 potential virulence factors of the strain, which provides a theoretical basis for subsequent studies on their pathogenic mechanisms and vaccine development for Mcc.

By comparative genome analysis of Mcc HN-B and the other six members within the Mm cluster, the core genome of the Mm cluster and unique families of each strain were identified. The phylogenetic tree and heatmap of ANI analysis revealed evolutionary and affinity relationships between the members of the Mm cluster. This provides a new perspective for understanding the key biological functions and main molecular characteristics of all members within the Mm cluster. Moreover, the core genome provides a reference for virulence research and drug development against the Mm cluster. The specific genes identified in each strain can be applied as new markers to diagnose and distinguish members of the Mm cluster.

Finally, the core genome of *Mycoplasma capricolum*, represented by Chinese isolates and 63 caprine-host-specific orthologs, was identified. These findings could provide material support for the pathogenic mechanism research, rapid diagnosis, and drug development for *Mycoplasma capricolum*.

## Figures and Tables

**Figure 1 microorganisms-10-02298-f001:**
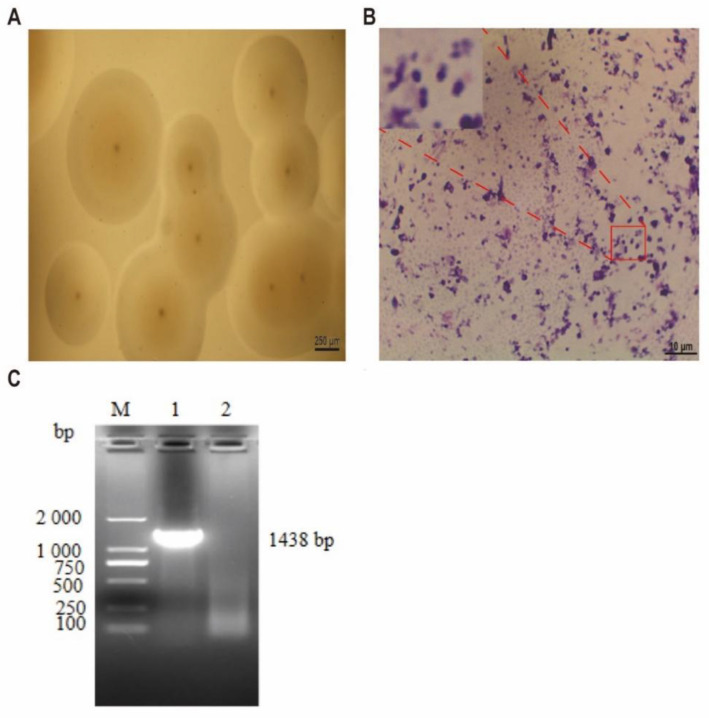
Mcc HN-B isolation and identification. (**A**) The morphology of the isolated strain, magnified 40 times (scale bar = 250 µm); (**B**) Giemsa staining results of the isolated strain, magnified 1000 times (scale bar = 10 µm): the top left picture results from the red-framed area magnified three times. (**C**) Agarose gel electrophoresis results of the *16S rRNA* gene: (M) D2000 Marker, (Line 1) the isolated strain, (Line 2) blank control.

**Figure 2 microorganisms-10-02298-f002:**
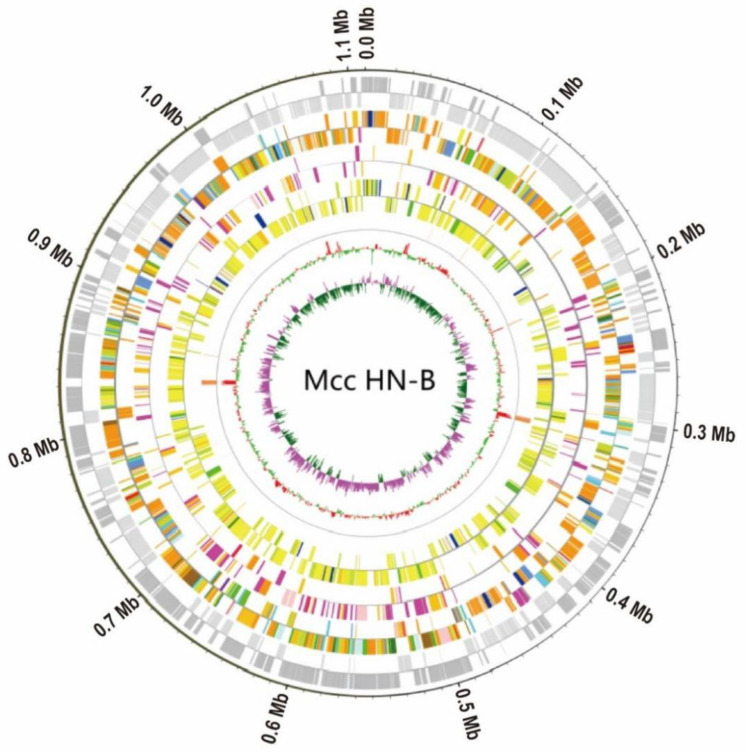
Whole-genome circle map of Mcc HN-B. The circles’ diagram from outside to inside is plus and minus strand gene (gray), plus and minus strand COG, plus and minus strand KEGG, plus and minus strand GO, ncRNA, GC content, and GC skew. KEGG uses the topmost category, represented by five colors. Genes that correspond to more than one category are assigned the sixth color. GC and GC skew are positive outward and negative inward, respectively. The window and step size are both 1000.

**Figure 3 microorganisms-10-02298-f003:**
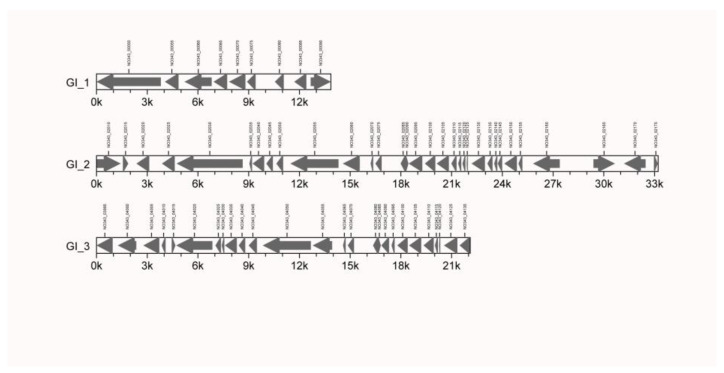
Gene island composition of Mcc HN-B. Note: The abscissa represents the length scale.

**Figure 4 microorganisms-10-02298-f004:**
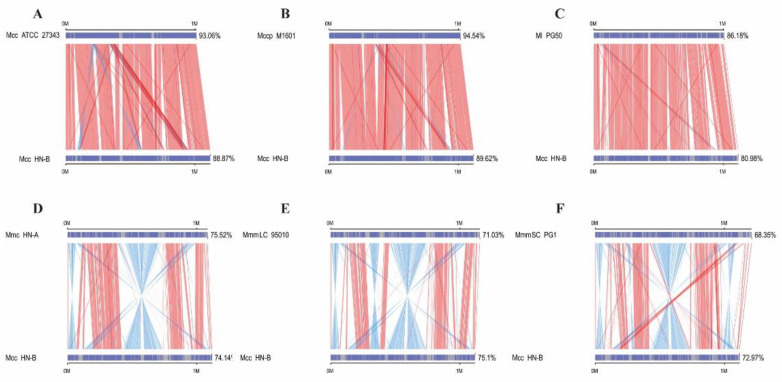
Parallel collinearity result plot. The upper axis represents the target species genome; the lower axis represents the Mcc HN-B genome. The red line indicates positive matching in the corresponding region; the blue line indicates reverse matching in the corresponding region.

**Figure 5 microorganisms-10-02298-f005:**
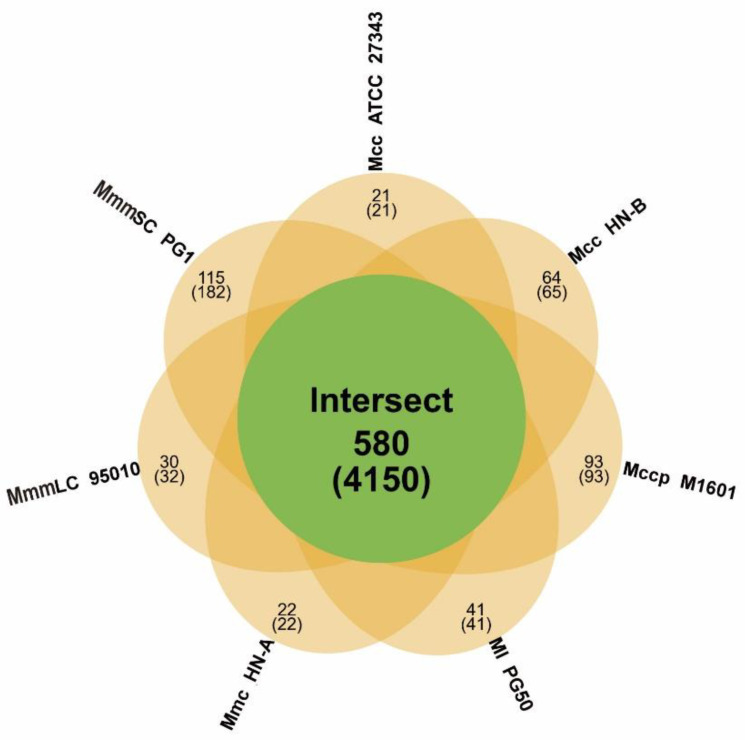
Homologous gene family analysis of the Mm cluster. An orange ellipse represents a genome. The green circle in the middle represents the intersecting part of all the ellipses. The figure above the bracket shows the number of gene families, and the figure in the bracket means the total number of genes in the corresponding gene family.

**Figure 6 microorganisms-10-02298-f006:**
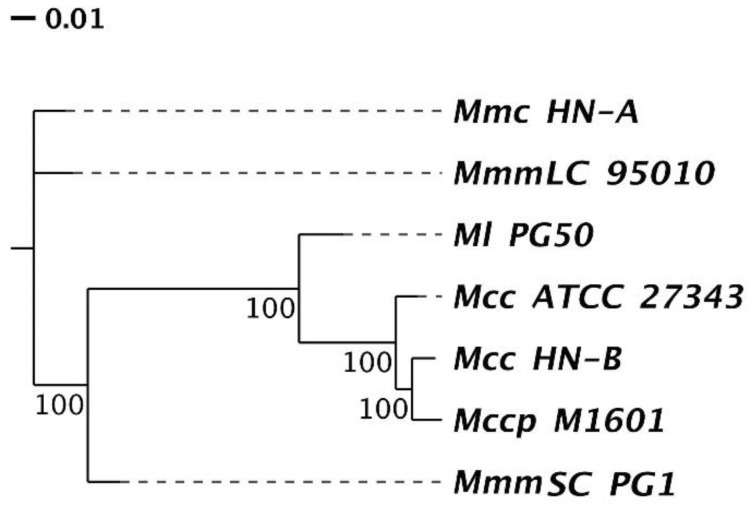
Phylogenetic tree based on the feature genes. The end of the branch represents different species. The node represents a common ancestor. The number on the branch shows its reliability. The branch length shows relative genetic distance within species.

**Figure 7 microorganisms-10-02298-f007:**
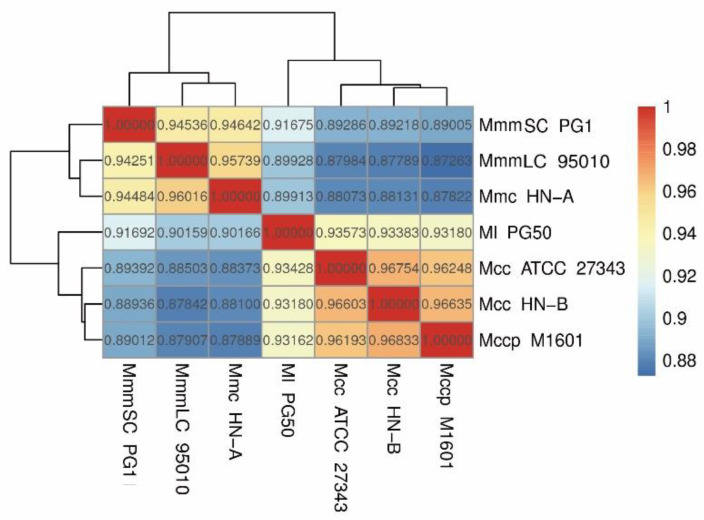
Heatmap of ANI analysis.

**Figure 8 microorganisms-10-02298-f008:**
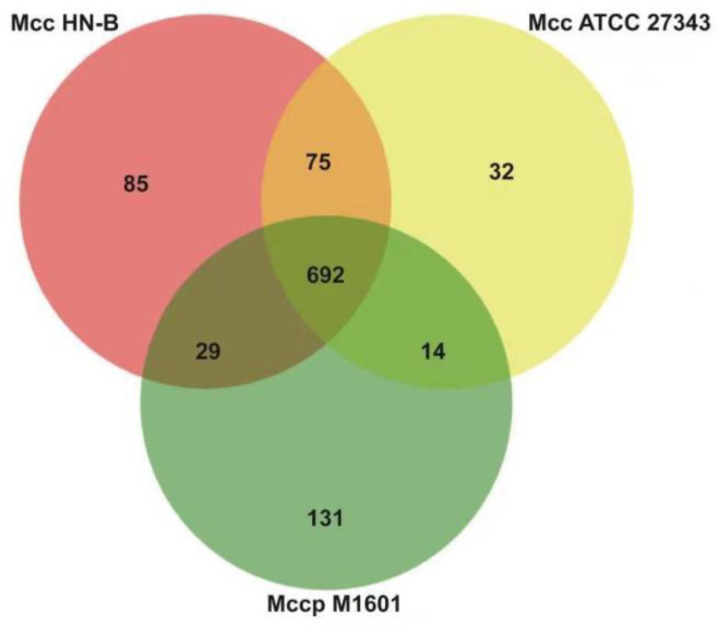
Venn diagram of orthologs in *Mycoplasma capricolum*. Note: Each circle represents orthologs of the strain.

**Figure 9 microorganisms-10-02298-f009:**
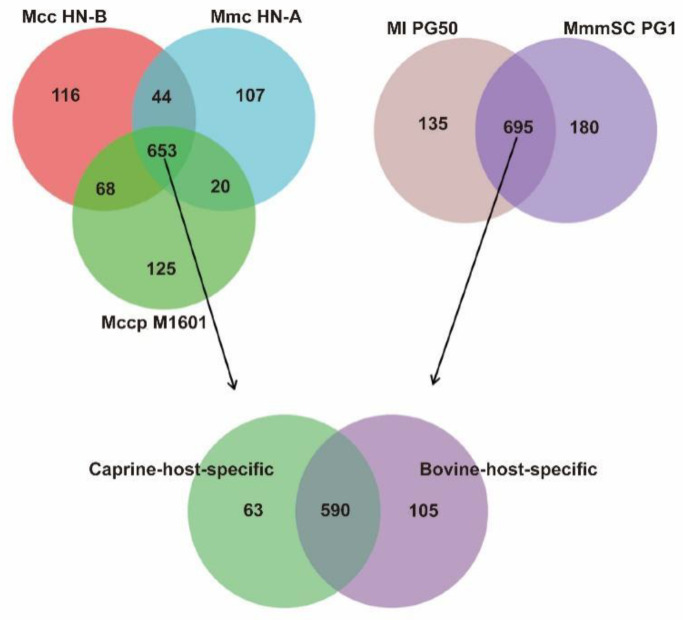
Venn diagram of caprine-host-specific orthologs. Note: Each circle represents orthologs of the strain.

**Table 1 microorganisms-10-02298-t001:** General characteristics of the Mcc HN-B genome.

Item	Number	Item	Number
Genome size (bp)	1,117,925	Number of rRNA genes	6
Genome GC content (%)	23.79	Number of tRNA genes	31
Number of genes	912	Number of sRNA genes	1
Length of gene (bp)	45–5,895	Number of interspersed repeats	6
Total length of gene (bp)	983,931	Number of short interspersed elements	4
Average length of gene (bp)	1078.87	Number of long interspersed repeated sequences	1
GC content of gene region (%)	23.95	Number of DNA elements	1
Total length of gene/genome (%)	88.01	Number of tandem repeats	175
Total length of intergenic region/genome (%)	11.99	Total bases in tandem repeats	20,145
GIs number	3	Number of prophages	0
Total GI length (bp)	69,172	Average length (bp)	23,057.33

**Table 2 microorganisms-10-02298-t002:** Basic information on each strain of the Mm cluster.

	Mcc HN-B	Mcc ATCC 27343	Mccp M1601	Mmc HN-A	MmmLC 95010	MmmSC PG1	Ml PG50
Accession number	CP093215	CP000123.1	CP017125.1	CP093215	FQ377874.1	BX293980.2	CP002108.1
Isolation place	China	USA	China	China	France	Sweden	USA
Host	goat	goat	goat	goat	goat	cattle	cattle
Collection date	2022	2005	2016	2021	1995	2003	2010
Size (bp)	1,117,925	1,010,023	1,016,707	1,084,691	1,153,998	1,211,703	1,008,951
G + C (%)	23.79	23.8	23.67	23.76	23.81	24.0	23.8

**Table 3 microorganisms-10-02298-t003:** Statistical results of SNPs, InDels, and SVs between Mcc HN-B and target strains.

	SNPs	Insertions	Deletions	SVs
Mcc ATCC 27343	17,885	500	497	98
Mccp M1601	19,302	634	652	104
Mmc HN-A	32,999	806	841	123
MmmLC 95010	33,126	849	912	122
MmmSC PG1	25,573	613	683	141
Ml PG50	33,196	703	792	97

**Table 4 microorganisms-10-02298-t004:** Gene family analysis results of the Mm cluster.

Species	Total Genes	Gene in Families	Unclustered Genes	Families	Unique Families
Mcc HN-B	912	849	63	818	64
Mcc ATCC 27343	825	804	21	792	21
Mccp M1601	889	796	93	773	93
Mmc HN-A	848	826	22	802	22
MmmLC 95010	927	899	28	837	30
MmmSC PG1	1,017	919	98	777	115
Ml PG50	850	809	41	789	41

**Table 5 microorganisms-10-02298-t005:** Specific genes to each strain within the Mm cluster.

Species	Total Genes	Shared Gene Number	Specific Gene Number
Mcc HN-B	912	847	65
Mcc ATCC 27343	825	804	21
Mccp M1601	889	796	93
Mmc HN-A	848	826	22
MmmLC 95010	927	895	32
MmmSC PG1	1,017	835	182
Ml PG50	850	809	41

## Data Availability

All the analyzed datasets in the current study are available from the corresponding author upon reasonable request.

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
