# Peer review of "First Report and Comparative Genomic Analysis of Mycoplasma capricolum subsp. capricolum HN-B in Hainan Island, China"

_microorganisms, 2022, doi:10.3390/microorganisms10112298_

Round 1
Reviewer 1 Report
The manuscript of Zhang and coworkers is an interesting work on comparative genomic analysis of Mcc HN-B which was firstly reported to isolate and identify in Haninan Island, China. The manuscript would gain on quality with respect to the following comments:
1. Typing errors should be carefully corrected.
2. The sentence “In addition, 1.5% agar powder……to prepare mycoplasma liquid medium” was unclear, even for the interested readers. The authors need to check this statement (line 90-91);
3. The virulence factors mentioned in this manuscript are all derived from automated computational analysis without experimental verification and lack of conviction. They can be expressed as potential virulence factors or putative virulence factors.
Author Response
The manuscript of Zhang and coworkers is an interesting work on comparative genomic analysis of Mcc HN-B which was firstly reported to isolate and identify in Haninan Island, China. The manuscript would gain on quality with respect to the following comments:
Q1- Typing errors should be carefully corrected.
Thank you for your suggestion, we have carried out a full-text review and correction. In addition,a Certificate of Editing was attached to the reply.
Q2-The sentence “In addition, 1.5% agar powder……to prepare mycoplasma liquid medium” was unclear, even for the interested readers. The authors need to check this statement (line 90-91);
Thank you for your careful suggestion, we have corrected it to "In addition, 1.5% agar powder (BioFroxx, Einhausen, Germany) was added to the mycoplasma liquid medium to prepare a mycoplasma solid medium."(line 251)
Q3-The virulence factors mentioned in this manuscript are all derived from automated computational analysis without experimental verification and lack of conviction. They can be expressed as potential virulence factors or putative virulence factors.
Thank you for your detailed suggestion, we have expressed the virulence factors mentioned in this article as potential virulence factors. We changed "14 virulence genes" to "14 potential virulence genes" and made changes in the appropriate place in the article.

Reviewer 2 Report
The information presented in the manuscript is essential, given that there is a lack of information in this regard.
However, the document must be written in English and verify that the verb tenses are used correctly. Sometimes, the times are mixed, and it is impossible to understand what the authors intend to convey. It is suggested that a native of the language review it.
On the other hand, the abbreviation for microliters must be spelled correctly and not as presented in the manuscript.
Authors should carefully check details such as writing polya instead of polyA (line 128). and these types of errors throughout the document. Correct this statement "the shortest protein sequence enjoys 40% amino acid similarity in 80% of its length (line 171)". Authors sometimes write Mycoplasma solid medium with capital letters and other times with lowercase letters. There needs to be consistency in writing throughout the manuscript.
Figure 1 can improve the quality of the information. For example, in panel A, they present it as the morphology of the isolated colonies; however, at this magnification and the type of photography, no morphology is appreciated. In all micrographies, the scale bar has to be added. In panel C the micrograph could be improved if a zoom of a particular region is shown.
The letters in figure 2 are not well observed due to their size. In general, the results should be better presented and explained.
Write pan-genome instead pangenome in all cases.
The phylogenetic tree requires an outgroup to further support the reconstruction.
Author Response
The information presented in the manuscript is essential, given that there is a lack of information in this regard.
Q1-However, the document must be written in English and verify that the verb tenses are used correctly. Sometimes, the times are mixed, and it is impossible to understand what the authors intend to convey. It is suggested that a native of the language review it.
Thank you for your helpful suggestions. We have invited native speakers of the language to review and correct them. A Certificate of Editing was attached to the reply.
Q2-On the other hand, the abbreviation for microlitersmust be spelled correctly and not as presented in the manuscript.
Thank you for your careful advice. We have corrected.
Q3-Authors should carefully check details such as writing polya instead of polyA (line 128). and these types of errors throughout the document. Correct this statement "the shortest protein sequence enjoys 40% amino acid similarity in 80% of its length (line 171)". Authors sometimes write Mycoplasma solid medium with capital letters and other times with lowercase letters. There needs to be consistency in writing throughout the manuscript.
Thank you for your careful suggestion, “Polya” has been corrected to “PolyA”. We have examined and corrected this type of error in full text. At the same time, we have uniformly used lowercase letters to express "mycoplasma solid medium". Finally, we correct "the shortest protein sequence enjoys 40% amino acid similarity in 80% of its length " to "80% of the shortest protein sequence has 40% amino acid similarity".(line 885)
Q4-Figure 1 can improve the quality of the information. For example, in panel A, they present it as the morphology of the isolated colonies; however, at this magnification and the type of photography, no morphology is appreciated. In all micrographies, the scale bar has to be added. In panel C the micrograph could be improved if a zoom of a particular region is shown.
Thank you for your constructive comments. We have improved the quality of Figure 1. Firstly, in panel A, we enlarged it locally to better show the colony morphology of Mcc HN-B on mycoplasma solid medium. Secondly, in panel B and panel C, we have added the scale bar respectively. Thirdly, the overall layout of the Figure 1 was adjusted for better display. Finally, we package the high definition image in Figure.zip for your viewing.
Q5-The letters in figure 2 are not well observed due to their size. In general, the results should be better presented and explained.
Dear reviewers, we have resized the images to better present the results. Since Figure 2 is a large high-definition image, we attach it to Figure. zip for your viewing.
Q6-Write pan-genome instead pangenome in all cases.
Thank you for your suggestion, we have carried out a full-text review and correction.
Q7-The phylogenetic tree requires an outgroup to further support the reconstruction.
Thanks for your constructive comments. The phylogenetic tree in this paper was constructed based on single-copy orthologs using ortholog results obtained from comparative genomic analysis. Mm cluster contains Mcc, Mccp, Mmc, MmmSC and other different species. Therefore, it shows the evolutionary relationship between different members of the Mm cluster cluster. For the Mcc HN-B strains we isolated and identified, other different species (Mccp, Mmc, MmmSC, etc.) can be used as outgroups to show their evolutionary relationship.
Round 2
Reviewer 2 Report
In the new version of the manuscript, the authors present satisfactory answers to what was requested by the reviewer. I consider that the quality of the writing has improved considerably with the modifications made.
Author Response
Comments and Suggestions for Authors
In the new version of the manuscript, the authors present satisfactory answers to what was requested by the reviewer. I consider that the quality of the writing has improved considerably with the modifications made.
Thank you for your approval of the manuscript. Your suggestion has helped us a lot.